# An Interpretable Multi-scale Deep Network for Structure Localization in Medical Images

Yongfei Liu, Bo Wan, and Xuming He

ShanghaiTech University, Shanghai China
{liuyf3, wanbo, hexm}@shanghaitech.edu.cn

**Abstract.** Anatomical structure localization plays an important role in image-based medical diagnosis. Despite recent progress achieved by deep learning methods, most existing neural networks for localization lack an interpretable inference process and thus are difficult to use for critical diagnosis in practice. In this paper, we propose an interpretable multi-scale convolutional networks for structure localization in medical images. Our network employs a modularized architecture consisting of a local branch to encode structure features and a context branch to capture global cues. The local branch adopts a coarse-to-fine strategy to refine the localization. And within each step, it learns a linear voting scheme based on a set of visual landmarks. The context branch uses a deformable pooling to encode contextual anatomical structures for reducing local ambiguities in localization. Given a prediction, we are able to trace back and determine which features are involved and their importance. We validate the proposed strategy on a Nuchal Translucency (NT) dataset, and the results demonstrate that our method is capable of generating an interpretable localization process and achieves the state-of-the-art detection performance.

## 1 Introduction

Localizing anatomical or abnormal structures is of great importance in medical image processing and its related diagnosis, such as pneumonia detection in chest radiographs [9], masses localization in mammograms [5] and lesion detection in 3D CT scan volumes [11]. Recently, thanks to rapid progress in deep learning, a promising strategy is to adopt an object detection network, such as R-CNN [8] or its variants, to directly predict the location of structures. While achieving strong performances, those deep networks are typically based on a complex and highly nonlinear functional mapping which is difficult to interpret. Such black-box like models are sensitive to input noises, and can generate false detection without much evidence on its decision process. As a result, it is difficult to adopt them in practice when structure localization is critical to the diagnosis.

There has been much effort on explaining deep networks for classification, which can be largely divided into two groups. The first group attempts to visualize learned filter patterns or activation maps based on gradient or sensitivity analysis [12]. For example, Zhao *et al.* [14] proposed respond-weighted class activation mapping to visualizing important input regions for classifying electron cryo-tomography images. The second type of work explains deep networks

by learning a "transparent" model that mimics the learned network. Hinton *et al.* [2] proposed to distill knowledge in deep networks into a decision tree, which maps the network inference into a sequence of simple feature-based decisions. By contrast, much less attention has been paid to those deep networks for object localization. Wu *et al.* [10] trained an And-Or Graph (AOG) to explain the classification of each proposal in the R-CNN [3], which however did not provide interpretation on how the object locations are determined by the network. Moreover, such an object-centric representation is unable to tackle the localization of fine-structures in medical images that heavily relies on global context cues.

In this work, we propose an interpretable deep learning strategy for structure localization to address above-mentioned limitations. Inspired by [13], our method adopts a feature-based voting scheme that allows us to represent the localization process by linearly combining predictions from multiple visual landmarks. In order to cope with noisy medical images, we develop a novel architecture consisting of two main components: a coarse-to-fine module that gradually refines the localization and learns a set of multi-scale visual landmarks and a global context module that captures the layout of surrounding anatomical structures and help reduce local ambiguities. Moreover, we design a simple trace-back step to uncover what features contribute to the final localization and their importance.

We validate our method on our private Nuchal Translucency (NT) Scan dataset. Results demonstrate superior performances and interpretability of our method. Our main contributions are two-folds: 1) We propose an interpretable voting-based deep network for structure localization task in medical images. 2) We develop a coarse-to-fine strategy and a multi-scale representation in voting to cope with noisy medical images.

## 2    Methodology

### 2.1    Method Overview

Given an image $I$, the task of structure localization aims to generate target location $\mathbf{x}$ and foreground confidence score $s$ of a specific anatomical structure. Here $\mathbf{x} \in \mathbb{R}^4$ denotes its bounding box parameters (center location $x, y$, width and height $w, h$). We first generate an initial set of region proposals $\{\mathbf{x}_p \in \mathbb{R}^4\}$. Then the main focus of this work is to design an explainable deep neural network to regress the structure location $\mathbf{x}$ based on each initial proposal $\mathbf{x}_p$ and predict its confidence score from the image cues.

To achieve this, we propose a multi-scale representation of the target structure and image context based on a set of learned visual primitives. It enables us to learn a sequential voting-based strategy to predict the structure location and its score. Hence by decomposing the prediction into multi-scale primitive estimation and voting, we provide an interpretation that identifies relevant image cues and a simple geometric rule that generate the outcomes.

Specifically, we develop a dual-branch deep network consisting of a *local branch* that encodes the target structure feature and a *context branch* that captures its global contextual cues. An overview of our model is illustrated in Fig. 1. Given an input image $I$ and a proposal $\mathbf{x}_p$, the local branch starts from a coarse

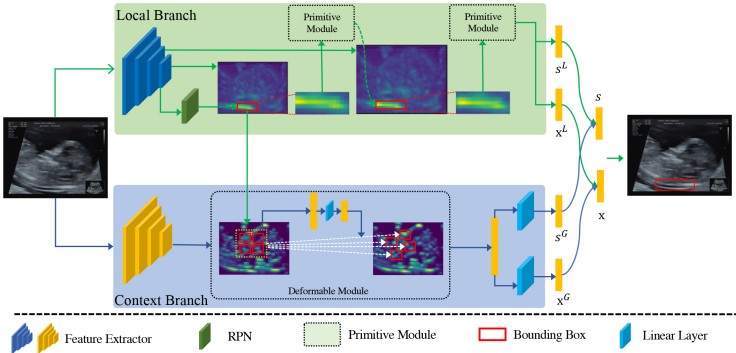

Fig. 1: **Overview of our framework.** Given a medical image, **Local Branch** can utilize local appearance feature and localize target anatomical structure in a transparent way. **Context Branch** captures global information to eliminate ambiguity. Finally, our model combines outputs in each branch to get bounding box and confidence score.

feature map within $\mathbf{x}_p$ and refines the location iteratively by gradually increasing the spatial resolution. Within each step, it employs a voting-based *primitive module* to generate a new location as a starting point for the next step. After $m$ steps, the local branch predicts a location $\mathbf{x}^L$ and a confidence score $s^L$. We use $m = 2$ as it typically saturates with more steps. The context branch uses a deformable convolutional module [1] to represent the global context and computes location $\mathbf{x}^G$ and confidence score $s^G$ from the convolution features. We form the final prediction by combining the outputs from both branches:

$$\mathbf{x} = \mu \mathbf{x}^L + (1 - \mu)\mathbf{x}^G, \quad s = \mu s^L + (1 - \mu)s^G \tag{1}$$

where $\mu$ is learned weight coefficient. Below we will introduce those two network branches in details.

### 2.2   Local Branch

In the local branch, we first use an ResNet-18 to compute a convolutional feature map $\mathbf{\Gamma}^L$ with $C$ channels, and train an RPN [8] to generate a set of proposals. For each initial proposal $\mathbf{x}_p$, we introduce a voting-based primitive module and apply it to the proposal in a coarse-to-fine manner to generate the branch prediction.

**Primitives Module.** Our primitive module generates a new localization $(\mathbf{x}^L, s^L)$ by taking a cropped conv-feature map around its input $\mathbf{x}_p$. Inspired by [13], we adopt a voting strategy to implement this module by introducing a set of visual primitives as localization landmarks. Formally, we introduce a set of visual primitives $\{\langle \mathbf{v}_k, \mathbf{D}_k \rangle\}, k \in \{1, \ldots K\}$, where $\mathbf{v}_k \in \mathbb{R}^C$, called *semantic template*, is a mid-level conv-feature pattern that represents a sub-part of anatomical structure. $\mathbf{D}_k \in \mathbb{R}^{M \times N}$, or *spatial template*, is a 2D heatmap that encodes average spatial relative distance between location of $\mathbf{v_k}$ and anatomical structure center.

In order to detect target structure, each primitive finds a corresponding area that has the highest response with $\mathbf{v}_k$, then votes to the center of target location

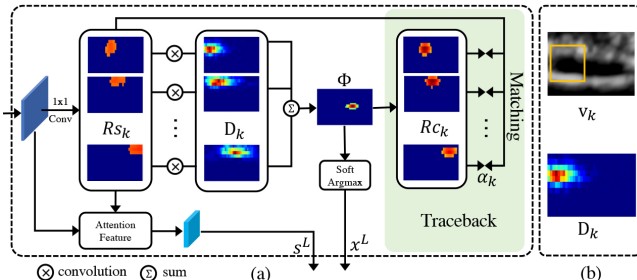

Fig. 2: **Primitive Module**: (a) demonstrates the voting process of visual primitives. (b) is an example of visual primitive.

with $\mathbf{D}_k$. Specifically, given the input conv-feature map $\mathbf{\Gamma}^L$ and $\mathbf{x}_p$, the module crops a structure-centric feature map $\mathbf{\Gamma}_p^L \in \mathbb{R}^{C \times M \times N}$ by applying ROI-Align and rescaling. Denote each $C$-dimensional feature vector in $\mathbf{\Gamma}_p^L$ as $\boldsymbol{\gamma}_i$ where $i$ is an index at the $M \times N$ grid and each feature vector is $\ell_2$-normalized so that $||\boldsymbol{\gamma}_i|| = 1$. We predict the target location in two steps:

*1) Primitive detection.* We estimate the location of each semantic template $\mathbf{v}_k$ within $\mathbf{x}_p$ by an inner-product based matching: $\mathbf{Rs}_k^i = max(\mathbf{v}_k \cdot \boldsymbol{\gamma}_i, 0)$, and $\mathbf{Rs}_k = [\mathbf{R}_k^i]_{M \times N} \in \mathbb{R}^{M \times N}$ represents the response map of each $\mathbf{v}_k$.

*2) Primitive voting.* We utilize each response map to vote for target anatomical structure center via spatial template $\mathbf{D}_k$, and accumulate all the votes for target location. We also use the response map as a spatial attention to reweigh the feature maps and predict the confidence score:

$$\mathbf{\Phi} = \frac{1}{K}\sum_{k=1}^{K}\mathbf{\Phi}_k = \frac{1}{K}\sum_{k=1}^{K}\mathbf{Rs}_k \otimes \mathbf{D}_k, \quad s^L = \mathcal{FC}^L(\sum_{k=1}^{K}\mathbf{Rs}_k \odot \mathbf{\Gamma}_p^L) \qquad (2)$$

where $\otimes$ denotes convolution and $\odot$ denotes element-wise product. $\mathbf{\Phi} \in \mathbb{R}^{M \times N}$ is a heatmap indicating the predicted center location, and $\mathcal{FC}^L$ represents multiple fully-connected layers. We then adopt differentiable soft-argmax [7] to convert $\mathbf{\Phi}$ into location $\mathbf{x}^L$.

After voting, we introduce a *trace-back* step to find which primitives contribute to the prediction as follows. First, we utilize the spatial template $\mathbf{D}_k$ to reconstruct each response map from the voting heatmap $\mathbf{\Phi}$, denoted as $\mathbf{Rc}_k \in \mathbb{R}^{M \times N}$. We then compute a matching score $\alpha_k$ between $\mathbf{Rs}_k$ and $\mathbf{Rc}_k$ to indicate the contribution of $\mathbf{v}_k$. Denote $\tilde{\mathbf{D}}_k$ as a flipped version of $\mathbf{D}_k$[1], our trace-back step can be written as

$$\mathbf{Rc}_k = \mathbf{\Phi} \otimes \tilde{\mathbf{D}}_k, \quad \alpha_k = \frac{\sum_i \mathbf{Rs}_k \odot \mathbf{B}_k + \sum_i \mathbf{Rc}_k \odot \mathbf{B}_k}{\sum_i \mathbf{Rs}_k + \sum_i \mathbf{Rc}_k} \qquad (3)$$

where $\mathbf{B}_k \in \mathbb{1}^{M \times N}$ is a binary mask, and $\mathsf{B}_k^i = 1$ when $\mathbf{Rs}_k^i > 0$ and $\mathbf{Rc}_k^i > 0$. The lager $\alpha_k$ indicates higher overlap area between $\mathbf{Rs}_k$ and $\mathbf{Rc}_k$ with high response value.

---

[1] As in 2D convolution, it can be formally written as $\tilde{\mathbf{D}}_k = \mathcal{J}_M \mathbf{D}_k \mathcal{J}_N$, where $\mathcal{J}_M \in \mathbb{R}^{M \times M}$, $\mathcal{J}_N \in \mathbb{R}^{N \times N}$ are anti-diagonal identity matrices.

**Coarse to Fine.** We propose to localize the target anatomical structure in a coarse-to-fine fashion by utilizing convolutional feature maps at multiple resolutions. Concretely, we first apply the primitive module to the *pool*-4 feature (downsampled 16 times), which updates the proposal $\mathbf{x}_p$ to $\mathbf{x}_p'$ at a coarse level. Then we use the primitive module on the *pool*-3 convolutional features (downsampled 8 times) within the updated proposal $\mathbf{x}_p'$. The final bounding box $\mathbf{x}^L$ and confidence score $s^L$ are the output from the fine-level voting.

### 2.3  Context Branch

In the context branch, we represent the global context of the target by a set of *context primitives*, which are used to generate a global prediction on the location $\mathbf{x}^G$ and confidence score $s^G$. We adopt the design of deformable pooling layer [1] so that the context primitives dynamically adapt to each input image.

Specifically, we use another ResNet-18 to compute feature map $\mathbf{\Gamma}^G$, and define a set of initial context proposals $\mathbf{x}_o \in \mathbb{R}^4, o \in \{1, \ldots, O\}$ to capture global context cues. $\mathbf{x}_o$'s are pre-defined regions in the neighborhood of $\mathbf{x}_p$ to cover a large portion of the input image. As in deformable RoI pooling [1], we define an union region $\mathbf{x}_u$ as the minimum box in spatial region that contains all $\mathbf{x}_o$, and union feature $\mathbf{\Gamma}_u^G$ are cropped from $\mathbf{\Gamma}^G$ with RoI-Align. The module computes a spatial translation $\boldsymbol{\delta}_o$ for each $\mathbf{x}_o$ by applying $\mathcal{FC}^{G_1}$ to the feature $\mathbf{\Gamma}_u^G$, and the context proposals are updated to $\mathbf{x}_o'$:

$$\boldsymbol{\delta}_o = \mathcal{FC}^{G_1}(\mathbf{\Gamma}_u^G) \qquad \mathbf{x}_o' = \mathbf{x}_o + \boldsymbol{\delta}_o \tag{4}$$

With the updated context proposals $\mathbf{x}_o'$, we crop all the context features $\mathbf{\Gamma}_o^G$ from $\mathbf{\Gamma}^G$ with RoI-Align, and concatenate them into a context representation $\mathbf{\Gamma}_{ctx}^G$. Finally, the location $\mathbf{x}^G$ and confidence score $s^G$ are predicted by $\mathcal{FC}^{G_2}$:

$$(\mathbf{x}^G, s^G) = \mathcal{FC}^{G_2}(\mathbf{\Gamma}_{ctx}^G) \tag{5}$$

## 3  Experiments and Results

**Dataset.** We validate our method on the private Nuchal Translucency Scan (NT) dataset in terms of localization performance and interpretability of its predictions. This dataset consists of ultrasound images from sonographic prenatal screening scans for detecting cardiovascular abnormalities in fetus. The scans are conducted during 11-14 weeks of pregnancy and are used to assess the quantity of fluid collecting within the nape of fetal neck. Our task is to localize nuchal translucency in ultrasound images for measuring its size as the chances of a chromosomal abnormality and mortality increase with thickening of the NT. This NT dataset contains 1073 subjects of image size $576 \times 768$ and we randomly split the dataset into three folds with two folds for training and one for test.

**Implementation Details.** The first three blocks of ResNet-18 [4] are taken to extract feature in local and context branch respectively. Then we set the number of primitives $K$=128, and the number of context proposals $O$=4. The local branch network is trained first, and then we train the whole network while finetuning the local branch with Adam optimizer. The initial learning rate is 1e-4 and decayed by a factor of 0.1 every 10 epochs.

Table 1: Localization performance on NT datast(%)

| Methods | Average | | |
|---------|---------|-----|-----|
| | AUC | SPE | SEN |
| Faster-RCNN | 83.6 | 91.2 | 81.4 |
| FPN | 83.9 | 91.1 | 82.2 |
| LS | 81.3 | 90.1 | 79.5 |
| LCF | 83.9 | 90.4 | 81.7 |
| **Proposed** | **85.6** | **92.0** | **84.3** |

Table 2: Results with various number of primitives.

| #Primitives | Average | | |
|-------------|---------|-----|-----|
| | AUC | SPE | SEN |
| $K = 32$ | 84.14 | 91.19 | 83.16 |
| $K = 64$ | 84.97 | 91.37 | 83.72 |
| $\boldsymbol{K = 128}$ | **85.6** | **92.0** | **84.3** |
| $K = 256$ | 85.31 | 90.40 | 83.41 |

Table 3: Results with various number of context proposals.

| #Context | Average | | |
|----------|---------|-----|-----|
| | AUC | SPE | SEN |
| $O = 2$ | 84.59 | 91.33 | 83.32 |
| $\boldsymbol{O = 4}$ | **85.6** | **92.0** | **84.3** |
| $O = 6$ | 85.52 | 91.83 | 84.04 |

**Localization Performance.** We first evaluate the localization performance based on the standard setting in [5] and use Area Under ROC (AUC) as metric. A predicted bounding box is true positive when its IOU with the ground-truth is great than 0.5. In addition, we compute the specificity (SPE) and sensitivity (SEN) of the localization using top-1 prediction: for each image, we only take the box with highest score as positive.Three-fold cross-validation strategy is adopted to report average performances.

We compare our method with the state-of-the-art detectors, Faster-RCNN [8], FPN [6] in Table 1. Our interpretable framework both outperforms Faster-RCNN and FPN with a sizeable margin and achieves performance gain of of **1.7%** on AUC, **0.9%** on SPE and **2.1%** on SEN compared with FPN specifically. These results demonstrate the effectiveness of the proposed framework in anatomical structure localization tasks, and show that an explainable and modularized network design can achieve the same or even higher level of performance as its "black-box" counterpart. To validate our network design, we also perform a series of ablation study to evaluate the effectiveness of different model components: i) Using single-scale voting at the coarse level in local branch as in [13] (called LS); ii) Using the local branch only (called LCF); iii) Our full model (called Proposed). In Table 1, we can see that our coarse-to-fine strategy and global context branch improve all three metrics consistently. LCF outperforms LS with **2.6%** on AUC, **0.3%** on SPE and **2.2%** on SEN, while the full model obtains significant improvements with **1.7%** on AUC, **1.6%** on SPE and **2.6%** on SEN compared with LCF. In addition, we also conduct ablative studies on the numbers of primitives $K$ and context proposals $O$. As shown in Table 2, the model reaches best performance with $K = 128$, indicating that the network becomes robust with more visual primitives. But the performance decreases when $K = 256$ because of brining more background noise primitives. In Table 3, for context proposals, the performance saturates at $O = 4$. However, it may insufficient to capture context features when $O = 2$.

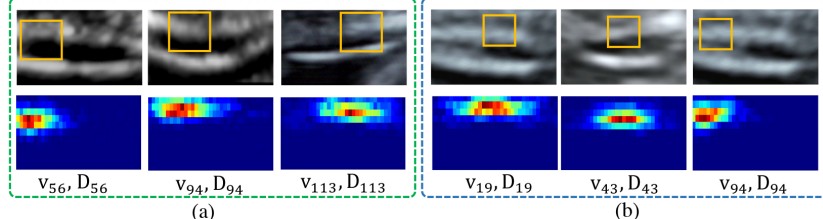

Fig. 3: **Interpretable Primitives.** Some representative primitives in coarse level and fine level of NT images are shown in (a) and (b). First row shows sub-parts (within yellow boxes) representing the receptive filed of semantic templates in origin image. Second row is the corresponding spatial templates, indicating the relative spatial location of the primitives w.r.t. target center.

**Model Interpretation.** We now demonstrate that our model predictions are easy to interpret by visualizing the inference procedure in the local branch and context branch. For the local branch, a few examples of primitives are shown in Fig. 3, which have clear semantic meanings. For instance, $\mathbf{v}_{56}$ is a semantic template for the sub-part on the left side of nuchal translucency (the yellow box region, denoted as $SP_{56}$), and $\mathbf{D}_{56}$ is its spatial template showing its relative location. We also illustrate the voting process in Fig. 4: (a) shows the intermediate localization result at the coarse level and fine level; (e) provides an example of the voting process at the coarse level. For instance, $\mathbf{Rs}_{56}$ indicates $SP_{56}$ exists in the middle area within proposal and $\mathbf{D}_{56}$ is its relative location w.r.t the ground-truth location (on the left side). Hence the primitive votes for moving to the right, as shown in $\mathbf{\Phi}_{56}$. In the trace-back process, we obtain $\mathbf{Rc}_{56}$ via back-projection as in Eq. 2 and Eq. 3. (c) shows the primitive weights which indicate the importance of each primitive during voting. At the fine level, the same inference process is shown in Fig. 4(f). In the context branch, our model can automatically capture informative context regions that indicate the global shape of a fetus, as shown in Fig. 4(b).

## 4  Conclusion

In this paper, we have proposed an interpretable deep framework for structure localization task, which consists of a local network branch for encoding structure features and a context branch for capturing global context features. Our method is capable of producing an easy-to-interpret localization process which highlights informative parts in the local region as well as surrounding context cues that help eliminate local ambiguity. In addition, we adopt a coarse-to-fine search to refine the target location so that our model can extract the different level of information and cope with noisy images. In the experimental evaluation, our method outperforms FPN, the state-of-the-art detector, with a sizable margin, and its prediction can be explained by tracing back through the intermediate stages.

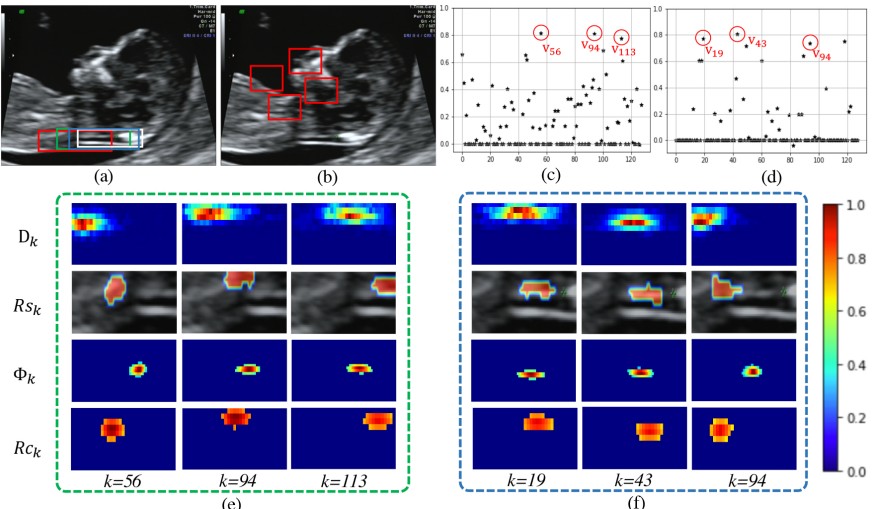

Fig. 4: **Voting Process.** (a) shows the bounding boxes generated in localization. The red box is the region proposal, green box is the localization result at the coarse level, blue box is the refined output, and white box is the final result. (b) shows context primitives in red boxes.(c)(e) are the weights of primitives and voting results of some representative primitives at the coarse level. (d)(f) are those results at the fine level.

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
