# OpenReview forum: "An Interpretable Multi-scale Deep Network for Structure Localization in Medical Images"
_MICCAI.org/2019/Workshop/COMPAY — Submitted to COMPAY 2019_

### Official Review · AnonReviewer1 · 2019-08-14
**A good idea that would benefit from being described in better detail.**

**Rating:** 6
**Confidence:** 4

**Review:**

In this paper, the authors attempt to create a new type of neural networks architecture consisting of two branches (local and contextual) and use the model in an interpretable way.
Globally, the paper presents a nice idea and attempts to solve the important problem of interpretability of neural networks. To be of general interest, the paper and methodology have to be clarified, freed from ambiguities and simplified. Also, method validation is incomplete.

Method :

•	The two branches form a rather complex model, and simpler versions could have been explored, for instance: using a single ResNet18 for feature extraction, not using deformable convolutions/pooling and merging both branches. These optimizations could simplify the model, decrease the number of parameters needed while approximatively keeping the same complexity and would simplify the explanations drastically.
•	The construction of the primitive module is rather complex and it would be beneficial to explain what decisions led to building it in this way. Also, in this module is defined a “matching score” that is not named, and followed by a long paragraph and equations to explain it. It would be simpler to describe this score as the DICE coefficient, and reword “matching score” as “similarity metric”.
•	The authors compare their new architecture with comparably old architectures (from 2014, 2015 and 2017) when newer and SOTA models exist, such as the Mask R-CNN (Kaiming He in 2017).
•	The authors experimented on a private dataset, reducing possibilities for future comparisons. The results would be more interesting if applied on a publicly available dataset. At the least, release of the dataset that was used would allow for the reproducibility of the paper.
•	The model interpretation paragraph failed to convince me of the interpretability of the so-called interpretable primitives. Both the semantic and spatial templates do not accurately encase the nuchal translucency, nor do they give much more information or understanding. Rephrasing the paragraph in a less mathematical way, more high-level, closer to the English language of a physician than to referencing the “templates” names would be beneficial.
•	The dataset is split in three parts, where two are used for training and one for test, but no independent validation set is used. This introduces a large risk of over training. For final results, averages of three-fold cross-validation are presented, but without reporting standard devotions. Significance of difference in performance is therefore not possible to judge.

language :

•	The English should be revised, as there are a few typographical mistakes and some words are being misused. For instance: “adopt” (Introduction, l. 5) and “black-box” (page 6) and “significant” (as it has a definition related to statistics) are misused, “lager” instead of “larger”, “filed” instead of “field”, “brining” instead “briging”.
•	The authors should define what interpretability means in the context of their paper.
•	The vocabulary needs to be restricted to a much smaller set of keywords, and there are many synonyms used along the paper to describe the same concept.
•	The authors need to define some words and concepts, such as: “visual primitives”, “context proposals”, “context primitives”, “global context cues”, “background noise primitives”.
•	In the paper, it seems that the concepts of “semantic template” and “spatial template” do not follow the definition of what a template is.
•	Page 5, the Coarse to Fine section needs some more explanations, what does pool-4 refer to? The 4th max-pooling layer of which model (RPN? ResNet18?)?
In the sentence “We use m = 2 as it typically saturates with more steps”, what does “it” refer to, and what is the meaning of saturate in this context?

---

### Official Review · AnonReviewer4 · 2019-08-15

**Rating:** 6
**Confidence:** 3

**Review:**

The paper proposes a deep two-branch architecture with better interpretability for structure localization and evaluates it on a private medical dataset.
The contribution is of interest, since interpretability is especially important in the medical domain.

The paper can be improved by
- explaining the local and context 'primitives' and the reasoning behind them in more detail
- evaluation on public datasets with comparison to other architecture's published results
- more examples and discussion of the interpetability of the primitives as this is the centerfold of the contribution
- evaluating how the algorithm scales with image size and number of training data
- source code would also be very helpful to better understand the architecture
- fixing minor language mistakes like in the two last sentences on page 6

Overall, the paper needs more investigation and discussion but offers an interesting contribution to the workshop.

---

### Official Review · AnonReviewer5 · 2019-08-15
**Out of Scope**

**Rating:** 1
**Confidence:** 5

**Review:**